# A Comprehensive Review of Tic Disorders in Children

**DOI:** 10.3390/jcm10112479

**Published:** 2021-06-03

**Authors:** Keisuke Ueda, Kevin J. Black

**Affiliations:** 1Department of Neurology, Washington University School of Medicine, St. Louis, MO 63110, USA; kevin@wustl.edu; 2Department of Psychiatry, Washington University School of Medicine, St. Louis, MO 63110, USA; 3Department of Radiology, Washington University School of Medicine, St. Louis, MO 63110, USA; 4Department of Neuroscience, Washington University School of Medicine, St. Louis, MO 63110, USA

**Keywords:** tics, Tourette syndrome, natural course, differential diagnosis, pediatric movement disorder, premonitory urge, comorbid symptoms

## Abstract

Tics are characterized by sudden, rapid, recurrent, nonrhythmic movement or vocalization, and are the most common movement disorders in children. Their onset is usually in childhood and tics often will diminish within one year. However, some of the tics can persist and cause various problems such as social embarrassment, physical discomfort, or emotional impairments, which could interfere with daily activities and school performance. Furthermore, tic disorders are frequently associated with comorbid neuropsychiatric symptoms, which can become more problematic than tic symptoms. Unfortunately, misunderstanding and misconceptions of tic disorders still exist among the general population. Understanding tic disorders and their comorbidities is important to deliver appropriate care to patients with tics. Several studies have been conducted to elucidate the clinical course, epidemiology, and pathophysiology of tics, but they are still not well understood. This article aims to provide an overview about tics and tic disorders, and recent findings on tic disorders including history, definition, diagnosis, epidemiology, etiology, diagnostic approach, comorbidities, treatment and management, and differential diagnosis.

## 1. Introduction

Movement disorders are central nervous system disorders that cause abnormal, unwanted movements and are usually unrelated to weakness or spasticity. Dysfunction of the basal ganglia and frontal cortex plays an important role in most movement disorders in children [1]. Conventionally, movement disorders are divided into two categories. The first category is hyperkinetic movement disorders, associated with an excess of movement (e.g., excessive, unnatural, and involuntary movement). These include tics, stereotypies, chorea, myoclonus, dystonia, and tremor [2]. The second group is hypokinetic movement disorders, with a paucity of movement (e.g., decreased amplitude, decreased speed, or loss of movement), including bradykinesia, akinesia, and rigidity [2]. Unlike hypokinetic movement disorders, hyperkinetic movement disorders, especially tic disorders, are relatively common in the pediatric population.

The most common movement disorders in the pediatric population are tic disorders, including Tourette syndrome (TS). In 1825, Jean-Marc Gaspard Itard reported the case of a French noblewoman who exhibited involuntary body movements involving the shoulders, neck, and face, and vocalization such as making barking sounds and uttering obscene language [3]. Subsequently, George Gilles de la Tourette referred to this case and reported nine patients with tic disorders [2]. Notably, these reports described the essential clinical features of tic disorders, such as early onset, waxing and waning course, echolalia, coprolalia, and echopraxia. For some time, a tic was regarded as a symptom of functional disorders like hysteria, neurosis, or narcissism. In 1968, the first case was reported of a patient whose tics improved with neuroleptics [4]. Since then, tic disorders have been discussed primarily in a neurobiological context. They are frequently accompanied by psychiatric comorbidities such as attention-deficit hyperactivity disorder (ADHD), obsessive-compulsive disorder (OCD), anxiety, and depression. Optimal treatment of tic disorders therefore requires a multidisciplinary approach involving neurologists, psychiatrists, psychologists, and behavioral therapists.

## 2. Definition

Tics are defined as “sudden, rapid, recurrent, nonrhythmic motor movement (motor tics) or vocalization (vocal or phonic tics)” [5]. Both motor and vocal tics are classified as simple or complex, although differentiating a simple tic from a complex tic is not always straightforward. Simple motor tics are brief, abrupt, repetitive, and seemingly non-purposeful movements, and involve only one muscle group or body part (e.g., face, neck, shoulders, or hands) [6]. Motor tics most frequently involve the eyes and mouth, followed by the neck and limbs; feet and midline axial structures are the least frequently involved [7,8]. Examples of motor tics include blinking, eye rolling, wide opening of the eyes or mouth, tilting the neck, raising the shoulders, and shaking the hands. Based on their phenomenology, simple motor tics are subdivided into three groups: clonic, dystonic, and tonic tics [9]. Clonic tics are abrupt, rapid, brief jerking movements (e.g., blinking, facial grimacing, head jerking). Dystonic tics are slower, resulting in briefly sustained abnormal postures (e.g., a prolonged involuntary upward deviation of the eyes, eye closure, bruxism, mouth opening, or torticollis). Tonic tics are isometric contractions (e.g., tensing of abdominal and limb muscles) [9,10,11]. Some tics may result in the transient interruption of ongoing motor activities or speech without loss of consciousness. Such tics are often referred to as blocking tics [12]. By contrast, complex motor tics are caused by several muscle groups and sometimes appear to be purposeful, coordinated, or orchestrated patterns of movement. Examples include touching, tapping, waving, kicking, jumping, echopraxia (mimicking others’ gestures), and copropraxia (performing obscene or forbidden gestures or inappropriate touching).

Simple vocal tics are meaningless sounds made by moving air through the nose, mouth, or throat. Vocal tics are often referred to as “phonic tics,” because the sound may be produced not only by contraction of the vocal cords but also by contraction of the nasal, oral, laryngeal, pharyngeal, and respiratory muscles. Examples include coughing, throat clearing, grunting, mimicking animal noises, and tongue clicking. Complex vocal tics involve several muscle groups and are characterized by words, phrases, or sentences. Examples include shouting and yelling, echolalia (repeating another person’s words), and coprolalia (uttering socially inappropriate expressions). Coprolalia is seen in 8–17% of patients with TS, and its onset is usually around the age of 15 years [13,14,15].

The Diagnostic and Statistical Manual of Mental Disorders, 5th edition (DSM-5), defines five tic disorders: Provisional Tic Disorder, Persistent (chronic) Motor or Vocal Tic Disorder, Tourette’s Disorder (also known as Tourette syndrome), Other Specified Tic Disorder, and Unspecified Tic Disorder [5]. The first three tic disorders require that the onset be before age 18 years and the symptoms not be caused by other medical illnesses such as Huntington’s disease, substance abuse, or medication side effects. Provisional Tic Disorder is considered when tics (motor or vocal or both) have been present for less than one year since tic onset. Both TS and Persistent (chronic) Motor or Vocal Tic Disorder indicate the presence of tics for longer than one year (though intervening tic-free periods are allowed). Persistent (chronic) Motor or Vocal Tic Disorder is diagnosed when individuals have exhibited either motor or vocal tics (but not both) at some time during the illness. When both motor and vocal tics have occurred during the course of the disease, though not necessarily concurrently, TS is diagnosed. Patients with TS tend to have a higher severity of tics, greater prevalence of complex motor tics (e.g., copropraxia and echopraxia), and more comorbid symptoms than patients with Persistent Motor Tic Disorder [16].

The final two tic disorders are new categories in the DSM-5. In the DSM-IV, tic disorders that did not meet the criteria for specific tic disorders were categorized as Tic Disorder Not Otherwise Specified. Instead of this category, the DSM-5 uses Other Specified Tic Disorder and Unspecified Tic Disorder when tic symptoms cause clinical distress or impairment, but the individual does not meet the criteria for the first three tic disorders. For example, Other Specified Tic Disorder is used when clinicians choose to describe the specific reason for not meeting the criteria (e.g., onset after 18 years), whereas Unspecified Tic Disorder is used when clinicians choose not to specify the reason why the criteria are not met (e.g., lack of sufficient information to make a more specific diagnosis). Importantly, the diagnostic criteria of tic disorders do not mention tic severity.

Under the DSM-IV, a diagnosis of Transient Tic Disorder required tics lasting for at least four weeks, and Persistent Tic Disorder and TS required a year of tics *without* a tic-free period of more than three consecutive months. These criteria were removed from the DSM-5. There was no factual basis for the four-week limit and most tic symptoms last longer than four weeks. The criterion of “three consecutive months” was not supported by epidemiological studies [17]. Furthermore, “Transient” Tic Disorder was changed to “Provisional” Tic Disorder in the DSM-5 due to confusion about the meaning of “transient” [Appendix 1 in ref. [18]]. Although some motor tics are stereotyped, the word “stereotyped” is no longer used in the definitions of tic disorders to avoid confusion between tics and Stereotypic Movement Disorder. In the DSM-IV, tic disorders had to cause clinically significant impairment or distress to patients, but this criterion was removed beginning with DSM-IV-TR. Tic symptoms do not always cause such impairment. Indeed, many children with tics are not always bothered by their tics, but rather by other neuropsychiatric comorbidities [19].

## 3. Clinical Course

In clinical practice, the diagnosis of tics is usually straightforward and can be made by reference to the patient’s clinical history. Although a complete physical examination with direct clinical observation is important in diagnosing neurological disorders, the neurological examination is normal in children with a tic disorder, except for the tics themselves [20]. Interestingly, patients with tics may subconsciously suppress their tics in the clinic [21]. In that case, it is wise to ask their guardians to video record their tics at home to understand their characteristics and severity. Diagnosis does not require laboratory testing or neurological imaging studies.

Understanding the natural course of tics is therefore essential. Tics usually begin between 3 and 8 years of age. The first tic symptoms are usually simple motor tics involving the face, head, or neck. Tics then spread in the rostro-caudal direction over time [22]. The first vocal tics occur on average a few years after the onset of motor tics and are usually simple vocal tics such as throat clearing or sniffing [22]. Conventionally, most tics are believed to go away on their own within a few months. However, epidemiological data for the clinical course of the new onset of tics are sparse. Conducting a follow-up study of Provisional Tic Disorder is difficult, as many patients do not seek medical help for their tics unless they find them bothersome [23]. A clinical follow-up study of children with recent onset of tics (less than six months since the first tic) showed that their tics persisted 12 months after the new onset, but most of the children were no longer bothered by their tics [24]. One explanation for this discrepancy is related to the benign nature of tic disorders. If tics are minimal or mild and do not cause any impairment or distress, patients and family members may not recognize them or feel it necessary to seek medical attention. If patients do not return to the clinic, this may be misinterpreted as remission of the tics.

If tics persist for more than one year, the severity usually peaks around 8–12 years of age [22]. Most patients with tics experience significant improvement or complete resolution by early adulthood [25]. Although tics are still present on direct examination in 88–100% of adults with a TS diagnosis in childhood or adolescence [26], about 33–47% of patients with TS report being completely tic free, less than 50% have mild tics, and less than 25% have moderate or severe tics in adulthood [22,25,26,27]. In other words, most patients with TS are no longer bothered by tics in adulthood. This fact can be welcome information for patients and guardians who are anxious about the prognosis of tic disorders [28].

Tics usually follow a waxing and waning pattern in severity and frequency, with a mixture of old and new tics. Although the onset of TS does not seem to be associated with life events, exacerbations and fluctuating severity of tics are related to environmental factors [29]. Tics may be temporarily exacerbated by psychological strains (e.g., stress, anxiety, excitement, anger), physical strains (e.g., fatigue, sleep deprivation, and infections), and environmental changes. This exacerbation is usually transient and subsides after the cause resolves. Individuals with tics often experience an unpleasant sensation preceding the tics, referred to as a premonitory urge, which is temporarily relieved by the execution of the tics [30,31,32]. Over 90% of adolescents and adults with TS [33,34,35] and 37% of children with TS from age 8 to 19 years [36] reportedly experience premonitory urges to some extent. Unlike other movement disorders, tics may be voluntarily suppressed for variable periods, which is known as tic suppressibility [11,37,38].

Various studies including genetic studies, neuropsychological testing studies, and neuroimaging studies have revealed only a few predictive factors of future tic disorders [28]. A clinical interview study of adults who had tics in childhood showed that tic severity in childhood and the presence of coprolalia were not necessarily predictors of adulthood tic severity [39]. In contrast, another clinical prospective longitudinal study of adults who had tics in childhood showed that severity of tics in late childhood was associated with tic severity in early adulthood [28]. Comorbid symptoms such as ADHD, OCD, and depression persist into adulthood and require close monitoring [40,41].

## 4. Epidemiology

It is difficult to estimate the true prevalence of tic disorders because a significant number of people do not recognize their tics or do not seek medical care [42,43]. The reported prevalence of tics in children therefore varies considerably [44,45,46]. Tics are more likely to affect boys than girls by a ratio of 1.5–4:1 [44,47,48]. The incidence of motor tics is higher in the winter months than in the spring months [47]. In a direct observational study where the researchers went to local schools to rate children’s tics, at least one motor tic was noted in 47% of 1st-grade students and 15% of 6th-grade students, and the overall cross-sectional prevalence of tics during childhood was approximately 19–24% [18,47,49]. Other observational studies using questionnaires revealed that tics were found in 22% of preschool children, 7.8% of elementary school children, and 3.4% of adolescents [50]. A similar tendency has been observed in other countries. For example, in Poland, the overall lifetime prevalence of tics is 9.9% (12% for boys and 7.7% for girls) [48]. In Spain, tics were noted in about 17% of school-aged students (19% for boys and 13% for girls) [51]. The prevalence of tics in children receiving special education was estimated to be 20–23%, with 5.3–7.0% of them meeting the criteria for TS [49,51].

The prevalence of TS is estimated at around 0.4–3% in regular school students [52,53,54,55]. Furthermore, the prevalence of TS in children with autistic spectrum disorder is higher: 22% of children with autistic spectrum disorder were found to have chronic motor tics and 11% were diagnosed with TS [56].

Tic disorders are the most common movement disorders in the pediatric population, but adult-onset primary tic disorder is rare [26,57] and is usually associated with underlying neuropsychiatric disorders (e.g., Down syndrome, Huntington’s disease, neuroacanthocytosis) [58,59], head or peripheral trauma [60,61,62], basal ganglia stroke [63], HIV infection [64], and neuroleptic and antiepileptic side effects or cocaine abuse [65,66,67,68,69]. Indeed, many adult tics are reoccurrences or continuations of childhood tics [39,70].

## 5. Etiology

The pathophysiology of tic disorders involves impaired function of cortical-striatal-thalamic-cortical circuits with aberrant associated neurotransmitter function, including dopamine, serotonin, and gamma-aminobutyric acid (GABA), and development-related atypical functional brain connectivity [31,71,72,73,74,75,76,77]. The etiology of tic disorders is complicated and multifactorial, including polygenic factors and non-genetic factors such as environmental factors and immune-mediated mechanisms, contributing to the heterogeneous clinical phenotype [78].

### 5.1. Environmental Factors

The specific environmental factors influencing the development of tic disorders have remained elusive. Various studies have investigated prenatal and perinatal epigenetic factors. A direct interview study found a correlation of tics with maternal life stress during pregnancy and nausea and vomiting during the first trimester of pregnancy [79]; a retrospective review study found a correlation with early prenatal care in the first trimester, more prenatal visits, decrease in the Apgar score at five minutes, and the month when prenatal care began [80]; and a systematic review found a correlation with low birth weight and prenatal maternal smoking [81,82]. However, a prospective, population-based, pre-birth cohort study identified alcohol use during pregnancy, cannabis use during pregnancy, and parity and inadequate weight gain during pregnancy as prenatal risk factors for chronic tic disorder, but other risk factors such as prenatal maternal smoking, gestational age, low birth weight, and complications of delivery were not associated with tic disorders [83].

In a case-control study [84], parental psychiatric disorders (especially in mothers) were associated with a greater likelihood of children developing TS and chronic tic disorders. A questionnaire survey showed that children from nuclear families with poor parental relationships were at more risk of developing TS [85]. However, the onset of tics was not associated with stressful life events [29].

### 5.2. Genetic Factors

In 1885, Georges Gilles de la Tourette pointed out possible genetic factors of tic disorders [86]. Since then, numerous studies have investigated genetic factors in tic disorders. Segregation analyses of 27 families with TS patients demonstrated autosomal dominant transmission patterns [87]. A twin study with probable monozygotic (MZ) and dizygotic (DZ) twins showed that concordance rates for tics were 77% and 23% for MZ and DZ pairs, respectively, and those for TS were 53% and 8% for MZ and DZ pairs, respectively [88]. A subsequent study with MZ twins also showed high concordance rates of 94% for tic disorders and 56% for TS, which indicates a genetic etiology with high penetrance [89].

Currently, tic disorders and TS are considered to be polygenic inherited disorders involving a large number of different genes. In a population-based cohort study using the Genome-wide Complex Trait Analysis program, the heritability of TS was estimated as 0.58–0.77 [90,91]. The same population-based study also revealed that the risk of developing tic disorders for first-degree relatives of probands with tic disorders was significantly higher than that for second and third-degree relatives [91]. Full siblings of individuals with tic disorders have a significantly higher risk of developing tic disorders than maternal half-siblings, regardless of similar environmental exposure, suggesting that the environment is much less important in causing tic disorders [91].

Several candidate susceptibility genes for TS have been suggested but have not been confirmed yet, probably because of the small sample size of each study and genetic and phenotypic heterogeneity [92]. A family with TS had an insertion or translocation of chromosomes 2 and 7 disrupting the *CNTNAP2* gene on chromosome 7q35-q36, probably leading to disturbed distribution in potassium channels in the nervous system [93]. Another case report of a family with TS showed that a deletion of exons 4, 5, and 6 of the *NLGN4* gene on chromosome Xp22.32-p22.31 was associated with TS and neuropsychiatric disorders such as autistic spectrum disorder, ADHD, learning disorders, anxiety, and depression [94]. The *SLITRK1* gene on chromosome 13q31.1, which encodes a single-pass transmembrane protein in the central nervous system, may be associated with TS as well as obsessive-compulsive disorders and schizophrenia [95,96]. The *HDC* gene on chromosome 15q21.2 encodes L-histidine decarboxylase that plays a role in histidine metabolism. A mutation in the *HDC* gene may be implicated in tics and TS by affecting histaminergic neurotransmission [97]. The *IMMP2L* gene on chromosome 7q31.1 encoding the inner mitochondrial membrane peptidase subunit 2 has also been implicated in TS [98,99]. In an animal study, the *Immp2l* mutation caused excessive mitochondrial superoxide generation and an increase in cellular oxidative stress [100]. However, a recent study using skin fibroblasts from patients with TS with the *IMMP2L* deletions revealed no evidence of substantial mitochondrial dysfunction in GTS fibroblasts [101].

A genome-wide screening of single nucleotide polymorphism (SNP) genotyping microarray with patients with TS showed 5 exon-affecting rare copy number variants involving the *NRXN1*, *AADAC*, *CTNNA3*, *FSCB*, *KCHE1-KCHE2-RCAN1* genes [102]. An analysis of a European ancestry sample of TS cases for rare (<1% frequency) copy number variants using SNP microarray data demonstrated that the *NRXN1* deletions and *CNTN6* duplications were associated with a substantially increased risk of developing TS [103]. Whole-exome sequencing studies identified the *CELSR3* gene on chromosome 3p21.31 [104] and the *ASH1L* gene on chromosome 1q22 [105] as high-risk genes for TS. Exome sequencing in patients with TS revealed possibly disrupting variants of the *OPRK1* gene on chromosome 8q11.23, encoding the opioid kappa receptor [106]. The Brainstorm Consortium genome-wide association studies (GWAS) of TS have indicated an association between TS and the *COL27A1* gene on chromosome 9q32-33 [107,108]. These results converge to the conclusion that tic disorders are not single-gene disorders, but genetic predisposition with environmental factors might increase the risk of developing tic disorders. Further studies are necessary to establish biological correlates of these genes and tic disorders.

### 5.3. Immunologic Factors

Abnormal immune responses have been proposed as underlying causes of tics, and such proposals have prompted investigation with different approaches, including animal studies, post-mortem studies, and laboratory studies. These studies have revealed some evidence for immune-mediated mechanisms, including impaired activation of the innate immune response, especially to bacterial infection [109]; elevation of interleukin-12 and tumor necrosis factor at baseline [110]; deficit in T-cell regulation [111]; increased titers of adhesion molecules as markers of a systemic inflammatory response [112]; altered immunoglobulin profile [113]; presence of oligoclonal bands in central spinal fluid [114,115,116]; microglial involvement; over-activity of systemic immune responses; and dysfunctional neural-immune crosstalk [117].

Some cases of TS may represent a central nervous system autoimmune disorder following infection. In 1978, a case was reported of an 11-year-old boy who developed severe tic symptoms after a febrile streptococcal infection, evidenced by elevated anti-streptolysin O (ASO) titers [118]. The patient did not respond to haloperidol but responded dramatically to prednisolone. In 1992, it was reported that two children developed tic symptoms after a streptococcal infection and also did not respond to neuroleptics but reached complete remission with adrenocorticotropic hormone and prednisone [119]. A clinical study of children with recent onset of movement disorders (tics and chorea) found that children with these movement disorders were more likely than those without to be positive for an anti-neuronal antibody directed against caudate nuclei and had elevated ASO titers than children without the movement disorders [120]. Subsequent case-control studies of patients with TS showed a significant increase in streptococcal antibodies such as ASO and anti-basal ganglia antibodies [121], and antibodies against putamen, but not the caudate or globus pallidus [122]. A large case-control study of 150 children with tic disorders demonstrated a significant elevation of ASO titers in children with tics compared to those of controls and a positive correlation between ASO titers and tic severity [123]. A cross-sectional study comparing children and adults with TS to children with other neurological diseases, children with recent uncomplicated streptococcal infections, adults with neurological disease, and healthy adults revealed a significant elevation of ASO titers in both children and adults with TS [124]. The most common basal ganglia binding antigen was similar to the proposed antigen in Sydenham’s chorea [124]. Thus, an autoimmune process caused by the cross-reacting of streptococcal bacterial antigens to the brain anti-neuronal antibodies, similar to the process of Sydenham’s chorea, was suggested as a pathophysiological mechanism of TS.

Pediatric autoimmune neuropsychiatric disorders associated with streptococcal infections (PANDAS) has been hypothesized as a subset of children with acute onset of tics or OCD following streptococcal infection. The diagnostic criteria include the following: (1) the presence of either OCD or tic disorder or both; (2) onset before puberty (usually between 3 and 12 years of age); (3) abrupt, dramatic, and explosive onset and/or episodic course; (4) a relapsing and remitting clinical course of symptoms with temporal relation associated with streptococcus infection; and (5) other neurological symptoms such as hyperactivity, anxiety, choreiform movements, or tics during exacerbations [125]. The pathophysiology of PANDAS is hypothesized to be related to IgG anti-neuronal autoantibodies produced through the process of molecular mimicry between host and group A streptococcus (GAS), which penetrate the blood–brain barrier and potentially induce neuronal signal transduction and subsequent excess dopamine release [126].

The findings of many studies, however, are inconsistent with the hypothesis of streptococcal infection inducing autoimmune-mediated antibodies to the basal ganglia and causing tic symptoms. In a case-control study of patients with TS, Sydenham chorea, autoimmune disorders, and healthy controls, IgG class anti-neural and anti-nuclear antibody titers were no longer significant when patients were stratified by age [127]. The same study also showed no significant difference in clinical symptoms depending on autoantibody positivity. Other methodologies such as enzyme-linked immunosorbent assay (ELISA) and Western immunoblotting demonstrated no total autoantibody abnormality to the caudate, putamen, and prefrontal cortex in patients with TS and PANDAS [128].

Multiple concerns about the diagnosis and treatment of PANDAS have been raised [129,130]. It is not straightforward to distinguish PANDAS from TS clinically, as they have several similarities. The onset of PANDAS overlaps with that of TS. Both PANDAS and TS have a relapsing and remitting clinical course. Symptoms of TS are typically worsened by physical and emotional stress, anxiety, or infection. A study using clinical interviews showed that the acute onset or exacerbation of tics was frequently seen in children with tic disorders (53%), which suggests that an explosive onset of tic symptoms is not unique to PANDAS [131]. Indeed, many children with PANDAS have pre-existing tics before the diagnosis of PANDAS, and so the abrupt, dramatic, and explosive onset of tics with the GAS infection may be the natural course of tic disorders. Patients with a diagnosis of presumed PANDAS may have a predisposition to tic disorders before the onset of PANDAS: first-degree relatives of children with PANDAS had a higher rate of OCD and TS than was reported in the general population [132].

Another concern for PANDAS is that no accurate diagnostic test exists to confirm or rule out PANDAS [133]. Diagnoses of PANDAS are often made based on incomplete criteria and most patients referred to specialty clinics with a pre-diagnosis of PANDAS do not fulfill the diagnostic criteria of PANDAS [134,135]. Making a diagnosis of a streptococcal infection is not always straightforward [136]. The sensitivity of a rapid streptococcal antigen detection test was low in children without clinical GAS symptoms such as tonsillar exudate or swelling, tender cervical lymphadenopathy, fever, and absence of a cough [137]. Even a positive throat culture may represent a GAS carrier state that does not require antibiotic treatment [138]. Anti-streptococcal antibody titers such as ASO and antideoxyribonuclease B need to be interpreted carefully. A prospective study revealed that 16% of children remained positive for the homologous serotype for more than 12 months after the infection, suggesting that single elevated antibody titers are often misleading and may only indicate antibody persistence from an infection that occurred a year earlier [139]. Single time-point cultures and single antibody titers are not sufficient to define the infection; sequential sampling of two different antibodies is necessary to make an accurate diagnosis of GAS infection [139]. As group C and G streptococci also produce antigenically identical ASO, the elevation of the ASO is not always linked to GAS infection [136,139].

Commercial antibody panel testing, such as the Cunningham Panel, is available for the diagnosis of PANDAS. This testing measures the human serum IgG level by ELISA directed against the dopamine D1 receptor, dopamine D2L receptor, lysoganglioside-GM1, and tubulin, and measures the activity of calcium/calmodulin-dependent protein kinase II [140]. However, the reliability of the Cunningham Panel is unclear. A case-control study to assess the diagnostic accuracy of the Cunningham Panel in patients with PANDAS or pediatric acute-onset neuropsychiatric syndrome (PANS), which is clinically defined by “the unusually abrupt onset of OCD and/or severe eating restrictions and at least two concomitant cognitive, behavioral, or neurological symptoms” [141], showed low sensitivities (15–60%), variable specificities (28–92%), and low positive predictive values (17–40%) and negative predictive values (44–74%) [142]. The same study also showed elevated CaMKII activity in 48% (10/21) of healthy controls and 66% (35/53) of patients with PANDAS or PANS. Moreover, at least one autoantibody titer was positive in 86% of the healthy controls compared to 92% of the patients. The European Multicentre Tics in Children Studies (EMTICS) study, a large longitudinal observational European multicenter project, investigated the role of environmental factors including GAS exposure in tic disorders [143]. The data from the EMTICS study failed to show evidence of specific neuronal surface antibodies or of GAS association with tic exacerbations in children with TS [144,145]. A prospective cerebrospinal fluid (CSF) analysis study with adults with TS also failed to demonstrate specific autoantibodies in the CSF [115].

The treatment of PANDAS is also controversial. The first-line treatment for acute and chronic PANDAS is a symptomatic approach using psychological and pharmacological interventions proven to benefit specific symptoms [146]. In light of the possible autoimmune mechanism, anti-inflammatory and immunomodulatory therapies (e.g., corticosteroids, non-steroidal anti-inflammatory drugs, intravenous immunoglobulins (IVIG), therapeutic plasma exchange, rituximab) have been used for the acute phase [147]. In double-blind placebo-control studies, however, IVIG did not significantly reduce tic severity or OCD symptoms [148,149]. In contrast, a single placebo-controlled study using IVIG and plasmapheresis for patients with severe, infection-triggered exacerbations of OCD or tic disorders, including TS, showed that plasmapheresis showed reduction of tic severity, as measured by the Tourette Syndrome Unified Rating Scale, but IVIG did not [150]. The Tourette Association of America concluded that “experimental treatments based on the autoimmune theory, such as plasma exchange, immunoglobulin therapy, or prophylactic antibiotic treatment, should not be undertaken outside of formal clinical trials” [151].

## 6. Comorbidities

Tourette syndrome is frequently comorbid with other psychiatric symptoms, such as ADHD, OCD, autistic spectrum disorder, depression, anxiety disorder, sleep disorders, migraine, and self-injurious behavior [152,153,154,155,156,157,158]. About 85–88% of patients with TS have at least one psychiatric comorbidity that usually appears between the ages of 4 and 10 years [159,160]. The most common comorbidities are ADHD and OCD. Other disorders (e.g., mood disorder, anxiety, disruptive behavior, self-injurious behavior) have been found to occur in about 30% of patients with TS [152,160]. Some of these may be part of TS (or share common etiological influences), whereas others may be less directly related. A review of this question is beyond the scope of the present review, but for instance, in the Brainstorm Consortium GWAS of 265,218 patients and 784,643 controls, TS shared a common variant genetic risk with ADHD, major depressive disorder, OCD, and migraine with aura [161]. Similarly, ADHD and OCD symptom clusters were genetically related to a diagnosis of TS [162].

Although TS and chronic tic disorders were associated with a higher risk of premature death, irrespective of the presence of comorbidities such as ADHD, OCD, and substance abuse, in a prospective cohort study [163], adults with TS reported good psychosocial functioning, attainment of social milestones such as graduating from school, securing a job, and getting married, and high quality of life [39,164]. As many people still have misconceptions about tic disorders (e.g., tics are due to psychological issues or individuals with tics cannot lead normal lives), providing anticipatory guidance at the time of the diagnosis of TS can reassure patients and their families [164].

Even when tic symptoms do not impair social, behavioral, or emotional functioning, comorbidities can negatively affect patients’ quality of life more than tics do [153,165,166]. Similarly, parents of children with tics reported that non-tic-related symptoms are more problematic than tics themselves [167]. A large clinical follow-up study of children and adolescents with TS showed an age-related decline in tics, ADHD, and OCD during adolescence, although other symptoms, such as sleep disturbance, remained [40]. Keeping in mind these comorbidities and their adverse consequences is important, and repeated assessment for them is warranted.

### 6.1. Attention-Deficit Hyperactivity Disorder

Attention-deficit hyperactivity disorder is characterized by inattentiveness, hyperactivity, and impulsivity that interferes with functioning or development. The average prevalence of ADHD in patients with TS is about 50–60% (within the range 33–91%), with male dominance [159,160,168,169]. The onset of ADHD symptoms usually precedes the onset of motor and vocal tics by an average of 2.4 years [160,168]. Disruptive behavior and functional impairment due to ADHD adversely influence academic, social, and family functioning [170]. Compared to pure TS, patients with TS and ADHD experience more deficits in planning, working memory, inhibitory function, and visual attention [171]. A heritability analysis involving individuals from affected sibling-pair families did not find significant genetic correlation between TS and ADHD, which suggests that high rates of ADHD in individuals with TS could be a result of increased but separate parental transmission of TS and ADHD susceptibility [172]. A case-control family study showed that ADHD and TS were not necessarily alternate manifestations of a single underlying genetic cause, but an increased risk of ADHD and TS in affected families may suggest an overlap in neurobiology and pathophysiology [173]. A GWAS also suggested modest shared etiological overlap between ADHD and TS [161].

The product labeling for stimulant medications recommends against their use in patients with tics and many clinicians still share this concern. However, substantial evidence has now accrued to dispel the idea that stimulants are contraindicated in children with tics [174,175]. An observational study of children and adolescents with ADHD found that new onset of tic disorder was actually less common in children treated with stimulants and tics remitted earlier in children who took stimulants [176]. A year-long, randomized, placebo-controlled trial of children with ADHD found the development of clinically significant tics to be equally common in children assigned to methylphenidate or placebo, and furthermore tics improved with treatment in two thirds of the children with a previous history of tics [177]. Most compellingly, a large, randomized, controlled trial involving children with TS and ADHD demonstrated a small but substantially significant improvement of tics with methylphenidate [178]. A meta-analysis concluded that methylphenidate did not worsen tic symptoms [179]. Additionally, a single dose of dexmethylphenidate transiently improved tic severity in a double-blind study [180]. Clinicians do observe occasional patients in whom a stimulant seems to induce or worsen tics. In the year-long study noted above, 23.6% of the children on an active drug developed moderate to severe tics for the first time—but so did 22.2% of the children on placebo! Thus, although it remains possible that methylphenidate may truly worsen tics in the occasional child, on average it is more likely to improve tics, and worsening with the drug is most likely to be coincidental. Thus, clinicians treating patients with tics and ADHD can use methylphenidate for the treatment of ADHD symptoms. However, given the warning in the product labeling, clinicians should explain the matter to caregivers when prescribing and monitor patients closely. Amphetamines have not been tested as carefully in tic patients, and in fact there is some evidence that children with tics tolerate methylphenidate better than amphetamine [181].

### 6.2. Obsessive-Compulsive Disorder

Obsessions are intrusive and unwanted images or thoughts that occur repetitively, and compulsions are behaviors that are performed to reduce the obsession or relieve obsession-related anxiety. The diagnosis of OCD requires OCD symptoms that are time-consuming (at least 1 h per day) or cause significant clinical distress or social or occupational functional impairment [5]. Lifetime prevalence rates of OCD in patients with TS are estimated to be 30–50% [160,182], whereas the general prevalence of OCD among adults is 1–3% [183,184]. The OCD symptoms in patients with TS usually begin within a few years after the onset of tics [160,185,186]. As TS and OCD can occur in clusters in families, a shared genetic architecture has been suggested, but a possible different underlying genetic susceptibility for TS and OCD compared with OCD alone has also been suggested [187,188].

Obsessive-compulsive disorder and TS have some common characteristics, such as a chronic waxing and waning course, premonitory phenomena preceding movement, and repetitive behavior [185,189]. Interestingly, obsessive-compulsive behaviors of TS differ somewhat from those of pure OCD. Contamination fears and negative thoughts (e.g., something going wrong, becoming sick, or something bad happening) are more prevalent in patients with pure OCD than in patients with TS and OCD [190]. In contrast, patients with TS tend to have compulsions such as counting, checking, ordering, arranging, touching, and hoarding as well as aggressive, sexual, religious, and symmetry obsessions [191]. Moreover, “just right” perception, where patients need to perform the same action repeatedly until they feel “just right,” is characteristic of TS with OCD [192]. The “just right” sensation can be distinguished from other premonitory urges in that the “just right” perception relates to a mental phenomenon (e.g., a want), whereas the premonitory urge involves a bodily sensation (e.g., an itch).

Trichotillomania is characterized by recurrent pulling out of hair and can be seen in both TS and OCD. Due to its repetitive nature, trichotillomania is generally considered part of the OCD spectrum, but trichotillomania has some common features with TS. Trichotillomania is typically preceded by urges but not obsessions and can be treated similarly to tics (e.g., antipsychotics and habit-reversal training) [193]. Similarly, skin picking is diagnosed separately in DSM-5, but has also long been considered a complex tic [194].

### 6.3. Anxiety and Depression

Anxiety and depression each occur in about 30% of patients with TS [160]. The high-risk age period begins around 4 years for anxiety disorders and around 7 years for mood disorders [160]. Comorbid depression positively correlates with tic severity [195]. Tourette syndrome patients with depression often have a positive family history of depression [195]. About 10% of youth with tic disorders experience suicidal thoughts and attempts, which often occur in the context of anger and frustration [196,197]. Although there is no correlation between suicidal ideation and tic severity, the presence of anxiety and depression increases the risk of suicidality in patients with tic disorders [196]. In a large epidemiological cohort study from the Swedish National Patient Register, adults with TS had about a four-fold higher risk of both suicide attempt and death [198]. It is therefore important to assess depression and anxiety symptoms, especially in patients with a positive family history of depression. 

### 6.4. Other Neuropsychological Symptoms

Eating disorders such as anorexia nervosa and bulimia nervosa are present in 2% of patients with TS, with female predominance and onset in adolescence (15–19 years old) [160]. Retching and vomiting may be symptomatic of tics if they occur alongside other tics and are accompanied by signs of tics such as suppressibility and premonitory urges [199]. However, medications for the treatment of TS symptoms, such as selective serotonin reuptake inhibitors and alpha-2 adrenergic agonists, can cause retching and vomiting [199]. Thus, clinicians caring for TS patients with gastrointestinal symptoms need to be aware of these possible causes.

Disruptive behaviors and rage episodes including explosive anger, temper outbursts, irritability, and aggressiveness are reported in 25–70% of TS patients [200]. The rage symptoms seem to be more closely correlated with the presence of other comorbid symptoms such as ADHD, OCD, and depression rather than tic severity [201]. Aggressive obsessions and compulsions, impulse dysregulation such as episodic rages, and risk-taking behaviors correlate with self-injurious behavior in TS [202]. Oppositional defiant disorder is reported in approximately 11–54% of TS patients and conduct disorder in approximately 6–20% [203].

Children with tics are at higher risk of being bullied and experiencing difficulties in socializing due to the social stigma [204]. Difficulties in school or the workplace often lead to discrimination, which may lower their self-esteem [204]. Poor self-perception and self-esteem are also related to the presence of comorbid psychiatric symptoms such as OCD, ADHD, and anxiety disorder [205]. The quality of life of patients with tic disorders is tied to their self-perception, so clinical treatment should focus on their self-concept and self-esteem [205]. Schizotypal personality disorder occurs in 15% of specialty clinic patients with TS and is associated with the presence of multiple psychiatric comorbidities [206]. Although patients with TS occasionally exhibit socially unacceptable behaviors, they rarely commit criminal acts [207].

### 6.5. Sleep Disorder

Sleep disorders occur in 64% of patients with TS [154]. Even after adjusting for potential confounding factors for sleep disorders, such as obesity, asthma, allergic rhinitis, anxiety, and depression, TS has been found to independently increase the risk of difficulty with sleep initiation and sleep maintenance, parasomnia, abnormal arousal, and excessive daytime sleepiness [154,208,209,210]. A polysomnographic study demonstrated that both motor and vocal tics are observable during all stages of sleep [208]. The likelihood of sleep disturbance is higher when ADHD or OCD is present [211,212]. Moreover, sleep disturbances themselves can aggravate tic symptoms in the daytime [211]. Treatment of sleep problems in patients with tics may therefore reduce the severity of tics as well as improve the sleep disorders themselves.

### 6.6. Headache

A headache is a common symptom in TS. A prospective questionnaire interview study showed that about 55% of children and adolescents with TS experience headache symptoms [213]. Migraine has been reported in about 17–27% of patients with TS with a mean age of 11.9 years [155,213,214]. This prevalence is higher than that of general school-aged children (2–10%) and adults (10–13%) [155,214]. Tension-type headache is also commonly seen in patients with tics. The interview study also reported that 28% of children and adolescents with TS had tension-type headaches, and the prevalence of tension-type headache is more than five times higher in tic patients than in the general pediatric population [213]. The exact mechanism of headaches in TS has not been elucidated, but a defect in serotonin metabolism has been suggested for migraine and tension-type headaches [155,213]. The GWAS showed significant positive correlations between migraine and TS [161].

### 6.7. Learning Disability

It is widely accepted that tic disorders do not affect intelligence and that most children with tics have normal or above normal intelligence. Although there is no evidence of memory or learning impairment in patients with TS [215], various factors such as tic severity, use of medication for tics, executive dysfunction, and coexisting ADHD, OCD, or other psychological illnesses can affect performance in school [216,217]. In a case-control study comparing children with TS only to children with TS and ADHD or highly suspected ADHD, learning disability was present in 23% of the children but only in those with TS with ADHD or highly suspected ADHD, not in those with TS only [218]. Patients with TS and tic disorders who seek treatment are more likely to underachieve academically across all educational levels, even after accounting for various confounders and comorbidities [219]. Detecting and addressing their difficulties at school (e.g., providing extra time to finish tasks, establishing a private space to release tics) is warranted to support their educational needs and enable them to reach their academic potential [219].

## 7. Treatment and Management

Given the frequently benign outcome and prognosis of tics, patients usually do not need to be referred to specialists. Adequate education for patients, family members, and schools is sufficient. If tics are severe and bothersome enough to affect quality of life, activities, or self-esteem, or cause significant social, emotional, and physical impairments, it is appropriate to refer the patient to specialists including pediatric neurologists, psychiatrists, psychologists, and behavioral therapists, depending on what problems most bother the patient.

Treatment and management of tic disorders begin with an assessment of tic frequency and severity and the presence of comorbid symptoms. To assess the frequency and severity of tic symptoms, the Yale Global Tic Severity Scale (YGTSS) is often used in clinics [220]. The YGTSS is a semi-structured clinical rating instrument for the assessment of tic severity in children, adolescents, and adults [194]. It is important to identify whether tics or comorbidities cause functional or emotional impairment. Unless tics are bothersome to patients, supportive care, reassurance, and education of the patient, family, and school are usually sufficient. The discussion should cover the diagnosis, the natural history of the disorder, activities or conditions that could worsen their tics, comorbidities, and indications for treatment. It is also important to debunk myths and misconceptions promulgated by stereotyped social media images. For example, many guardians and patients believe that TS is caused by stress or neglect; that tics will always progress to non-stop motor tics and socially inappropriate behaviors such as cursing; and that individuals with tics are intellectually impaired and cannot lead normal lives. In the U.S., the Tourette Association of America, a non-profit voluntary organization, and the Centers for Diseases Control and Prevention provide comprehensive materials to guardians and patients about TS and tic disorders in a variety of languages.

Pharmacological tic-suppressing treatment and behavioral therapy should be considered when tics cause physical, emotional, or social impairment (e.g., musculoskeletal injury, peer relation difficulty such as bullying, disruptive tic behaviors, low self-esteem, or difficulty in conducting physical or academic activities). The goal of treatment is to lessen the severity of the tics to improve the patient’s quality of life.

### 7.1. Behavioral Therapy

The American Academy of Neurology practice guideline recommendations summary mentions a form of behavioral therapy as the first-line treatment for tics [221]. Tic-suppression-based behavioral interventions consist of exposure and response prevention and habit-reversal therapy or its descendant comprehensive behavioral interventions for tics (CBIT) [222]. These two forms of training can be effective for both motor and vocal tics. Exposure and response prevention consists of repeated, prolonged exposure to stimuli that tend to induce tics and practice to resist tic behavior [223]. Theoretically, patients habituate to the unpleasant tic-provoking sensations, thus resulting in tic reduction [224]. Habit-reversal therapy consists of awareness training and competing-response training to encourage tic suppression for long periods of time [225]. Awareness training consists of self-monitoring tics and identifying early signs or warning signs, such as the premonitory urge. Competing-response training involves engagement in an active voluntary response that is incompatible with tics, such as tensing muscles antagonistic to tic-related muscles [225,226]. In clinical practice, patients and therapists first determine a tic hierarchy from most to least distressing and then first address the most distressing tic [225]. With practice, patients will be able to perform the competing response more effectively and efficiently [225]. A multicenter randomized controlled study of children and adults with TS comparing CBIT to psychoeducation and supportive therapy (PST) was conducted to analyze potential moderators for CBIT vs. PST or predictors of outcome [227]. Although both participants on tic medication and those not on tic medication showed response to CBIT, those on tic medication also reduced tic severity after PST. The study also showed that the presence of comorbid ADHD, OCD, or anxiety disorders, age, sex, “family functioning, tic characteristics, and treatment expectancy did not moderate response” to CBIT. A clinical study of children with tic disorders who underwent CBIT suggested that a positive response to CBIT was associated with improvement in anxiety, disruptive behavior, family strain, and social functioning at 6 months after treatment [228]. In a randomized controlled study of adolescents and adults with tics, those who received CBIT had less severe tics on the YGTSS than those who received PST at 6–8 years follow up [229].

The most widely accepted behavioral therapy for tic disorders is CBIT, which consists of habit reversal therapy, psychoeducation, functional intervention, and relaxation training [221,230]. It is safe, has no known adverse effects, and can be used by children and adults to help to reduce the severity of tics [225,231]. Although these therapies can eventually diminish the urges and decrease tic frequency and severity, finding a trained therapist can be difficult. Recently, Internet-based training programs have been made available to solve this issue. For example, “TicHelper” is an interactive, self-administered online CBIT program for school-age children. It is an 8-week program designed to teach tic management skills where children and guardians can learn about tics, such as how to reduce tic triggers and employ tic-blocking techniques [232]. “TicTrainer” is a free but untested Internet-based tic suppression training program consisting of a reward-enhanced exposure and response prevention strategy. Users attempt to suppress their tics to gain rewards, as if they were playing a video game. For example, they earn points for tic suppression and the longer they suppress their tics, the more points they earn [233]. Internet-based behavioral therapy gives patients and guardians more access to evidence-based therapies and may be effective in the long run [234]. Furthermore, group-based behavioral therapies have been developed to provide the behavioral therapies for more patients [235]. It likely will prove more cost-effective than individual therapies, and provides an opportunity for patients to meet other patients and to share their experiences and support.

### 7.2. Other Non-Pharmacological Treatment

Various complementary and alternative medicines have been used for the treatment of tics, including prayer, vitamins, massage, dietary or nutritional supplements (e.g., the B vitamins, vitamins C, D, and E, calcium, magnesium, Coenzyme Q10, fish oil), chiropractic manipulations, meditation, diet, yoga, acupuncture, hypnosis, homeopathy, and biofeedback [236,237]. Acupuncture may be effective for the treatment of tics for a short period, but the evidence is limited due to biases [238]. Without randomized controlled studies to establish dosing, safety, and efficacy, we do not know which of these have specific benefit beyond any placebo effect [237]. Since patients often pursue complementary and alternative medicines without informing their physicians, the treating physician needs to ask about these therapies [236].

### 7.3. Pharmacological Treatment

Pharmacological treatment should be considered when behavioral interventions fail or are not available (e.g., lack of access to behavioral therapies, including expense, or patient factors such as age, cognition, or willingness to participate) or when patients exhibit severe, violent tics that need immediate treatment. Before starting patients on a course of medication, clinicians must explain the purpose and set a realistic goal—that is, to reduce the severity and frequency of tics to the extent that they no longer bother the patient or cause significant problems.

Various therapeutic algorithms have helped clinicians choose medicines for the treatment of tics. One approach is to focus solely on efficacy, as the recent review of evidence from the American Academy of Neurology does [239]. However, in clinical practice, other factors, including side effect risk, cost, and convenience also play a role [221]. For instance, many families are more concerned about risk than about immediate benefit. Thus, we and several other authors prefer an approach that includes two tiers of medicines based on tic severity [240,241,242]. The first-tier medicines are for mild tics, when tolerability may be more important than proven efficacy, and the second-tier medicines are for severe tics or tics that are resistant to the first-tier medicines. As a rule, the medicines should be started on a low dose with gradual titration until they become effective. Once the symptoms subside to such a degree that the tics are no longer bothersome, physicians should discuss weaning off the medicines with patients and caregivers. The trial of weaning off the medicines should be undertaken only when the patient is mentally and physically healthy and does not anticipate any stressful situations or events.

#### 7.3.1. First-Tier Medicines

Medicines in this category are non-dopaminergic agents that are mildly to moderately effective in tic suppression and do not have severe adverse effects. The typical medicines in this category are alpha-2-adrenergic agonists such as clonidine and guanfacine [243]. As the alpha-2-adrenergic agonists are useful for the treatment of ADHD in children and adolescents [244,245], both tics and ADHD symptoms may improve. Clonidine is frequently used for the treatment of tics. The Tourette’s Syndrome Study Group conducted a large randomized controlled trial involving children with TS and ADHD to investigate the efficacy of clonidine and methylphenidate for tics and ADHD [178]. The children who took both clonidine and methylphenidate showed the most improvement, followed by those who took only clonidine. In addition to an oral preparation of clonidine, an adhesive transdermal patch is available. A randomized controlled study of children with tic disorders showed its efficacy and safety in tic treatment [246]. The recommended starting dose of clonidine is 0.025–0.05 mg/day, and it should be slowly titrated up to a therapeutic range of 0.1–0.3 mg/day [247,248]. The maximum total daily dose is 0.4 mg/day, divided up to 4 times a day with the highest single dose being 0.2 mg [249]. The side effects of clonidine include sedation, drowsiness, lightheadedness, tiredness, irritability, dry mouth, bradycardia, and hypotension. Although electrocardiogram monitoring is not necessary [250], close monitoring of blood pressure and heart rate is essential. Due to the side effect of sleepiness, clonidine may be useful for patients with tics who have difficulty with sleep initiation.

Guanfacine is another alpha-2 adrenergic agonist that has been used for the treatment of high blood pressure and ADHD. Its therapeutic effect for tics has been reported in open-label studies of children with TS and ADHD [251,252] and a placebo-controlled clinical trial of children with TS and ADHD [253]. The adverse effects are similar to those of clonidine, such as fatigue, drowsiness, dry mouth, headache, and irritability [254]. Guanfacine acts on alpha-2 receptors of neural cells more selectively than clonidine does, and it has a longer half-life and less severe adverse effects such as sedation and dizziness [255]. Thus, guanfacine is often more favored than clonidine. However, a randomized double-blind study of the extended-release formulation of guanfacine in children with moderate-to-severe chronic tic disorder did not show a clinically meaningful effect for tic suppression [254]. The recommended starting dose of guanfacine is 0.5–1.0 mg/day, and it should be titrated slowly up to a therapeutic range of 1.0–4.0 mg/day [247,248].

Based on a hypothesis that inhibitory GABA signaling dysfunction in the basal ganglia underlies the pathology of TS [256], GABAergic medications such as antiepileptic drugs and baclofen have been used for the treatment of tics. Evidence of the efficacy of these medicines is not as robust as that of alpha-2 adrenergic agonists, but these medicines can be used as a monotherapy or as add-on therapy in the treatment for tics.

Topiramate is a broad-spectrum antiepileptic drug that enhances GABAergic activity and inhibits kainate/AMPA glutamate receptors [257]. An open-labeled study and a randomized double-blind study of children and adolescents with TS showed moderate efficacy and a statistically significant reduction in the YGTSS [258,259]. Adverse effects included headache, diarrhea, abdominal pain, drowsiness, cognitive slowing, and kidney stones [258]. The starting dose is 25 mg/day, and it should be titrated up gradually to 50–200 mg/day depending on efficacy and tolerability [258].

Levetiracetam is an antiepileptic medicine that enhances GABA inhibition of neuronal circuits [260]. Open-label studies of children and adolescents with TS showed statistically significant improvement of tic symptoms [261,262]. Their behavioral and school performance also improved with levetiracetam [261]. On the other hand, randomized double-blind studies of children and adults with TS showed no reduction in tic severity or improvement in behavioral scores [263,264]. Adverse reactions include irritability and somnolence. The initial dose of levetiracetam is 20 mg/kg/day or 250 mg/day [261,262]. The dose is increased gradually up to 30–40 mg/kg/day, with a maximum dose of 60 mg/kg/day or 2000 mg/day [261,262].

Clonazepam is a benzodiazepine that has been used for the treatment of seizures and panic attacks. It acts on benzodiazepine receptors, increasing the effect of GABAergic transmission [265]. A case report of a 13-year-old boy with TS showed that his tics improved with clonazepam [266]. A single-blind study of patients with TS who had high ratios of red blood cell to plasma choline levels showed more response with clonazepam than haloperidol [267].

Baclofen is a GABA B receptor agonist and has been used for the treatment of spasticity. A large cohort open-label study of children with TS revealed a significant reduction in the severity of tics [268]. A double-blind, placebo-controlled crossover trial of children with TS demonstrated a reduction in tic severity, as measured by the YGTSS, although the reduction was not statistically significant [269]. Common side-effects are sedation and drowsiness [268]. The starting dose is 10 mg/day, and it can be increased gradually by 10 mg/day weekly, depending on the patient’s symptoms, up to 80 mg/day (mean 30 mg/day) [268].

#### 7.3.2. Second-Tier Medicines

Medicines in this category are dopamine receptor blocking agents (DRBA) (e.g., typical and atypical neuroleptics). Abnormal dopaminergic activity is considered to be involved in the mechanism of TS [270]. Specifically, D2 receptor blocking agents reduce tic severity by about 70% [248]. They are more effective in tic suppression than the first-tier medicines but cause more adverse effects such as sedation, metabolic syndrome (e.g., obesity, insulin resistance, hypertension, and dyslipidemia), akathisia, dystonia, or rigidity-bradykinesia [271,272].

Cardiovascular adverse effects such as torsades de pointes, QTc prolongation, myocarditis, and cardiomyopathy can be life-threatening [273,274]. Antipsychotic-induced tardive dyskinesia (TD) is also common in schizophrenia affecting approximately 21% of patients with schizophrenia receiving atypical neuroleptics and 30% receiving typical neuroleptics [275]. However, TD is apparently quite rare in TS. The classic symptoms of TD are involuntary, repetitive hyperkinetic movement around the mouth (e.g., chewing, protrusion of the tongue, jaw movements, lip-smacking, or puckering) [276]. The dyskinesia usually emerges insidiously and can occur after short-term or long-term use of neuroleptics [277]. TD can be distinguished from tics by the time course and by several other features, including that TD is usually rhythmic at 0.5–2 Hz, does not include premonitory phenomena, and is perceived by the patient as truly involuntary. The DSM-5 describes TD as medication-induced dyskinesia that occurs in the following conditions: (1) during exposure to a DRBA for at least three months, or one month in patients aged 60 years or older; (2) within four weeks of withdrawal from an oral medicine or within eight weeks from a depot medication. The symptoms should persist for at least one month after discontinuation of an offending agent [5]. Some experts argue that TD can occur within one year after exposure to DRBAs [277]. By definition, TD symptoms persist for at least one month after the discontinuation of medicine [278]. A rapid reduction or sudden discontinuation of the medicine may lead to transient worsening of dyskinesia, known as withdrawal dyskinesia [275,279]. Slow and gradual tapering of the offending medication over a period of weeks or months is recommended for patients who develop TD and do not have a psychiatric illness [280], though limited evidence supports slow tapering vs. abrupt discontinuation. Although many patients with TD showed some improvement within the first year of discontinuation of the neuroleptic, the chance of complete remission has been reported to be low in both psychiatric and non-psychiatric patients [281,282].

The medications approved by the U.S. Food and Drug Administration (FDA) for the treatment of tics are haloperidol, pimozide, and aripiprazole. Haloperidol is a D2 receptor antagonist and was reported to be effective for tic treatment in the early 1960s [283,284]. A case series [285] and a double-blind placebo-controlled study [286] demonstrated a similar therapeutic effect. A double-blind crossover study of haloperidol, pimozide, and a placebo revealed that haloperidol was slightly more effective than pimozide [287]. However, in a placebo-controlled double crossover study of children and adolescents with TS, haloperidol failed to provide significant benefit in tic reduction and was inferior to pimozide in tic suppression [288]. Furthermore, haloperidol causes more lethargy and extrapyramidal side-effects than pimozide [286,288,289]. Haloperidol is therefore not regarded as a preferred treatment for tic suppression but instead is recommended for patients who have not responded to other tic-suppressing medicines [290]. The starting dose is usually 0.5 mg/day, and it should be titrated gradually by 0.25–0.5 mg every 5–7 days. The typical dose ranges from 2 mg to 10 mg daily [290].

Pimozide is a potent DRBA and has been reported to be effective in tic suppression in clinical studies [291,292,293,294]. A prospective and double-blind clinical cohort study of children with TS demonstrated that pimozide was effective in the short and long term [295]. Arrhythmias such as QT prolongation and decreased blood pressure have been associated with patients receiving pimozide [287,296,297,298]. The starting dose is usually 0.05 mg/kg daily, and it should be gradually increased to 0.2 mg/kg, not exceeding 10 mg/day [299]. Because coadministration of potent cytochrome P450 (CYP) 2D6 inhibitor (e.g., sertraline) can increase pimozide concentration [300] and the arrhythmogenic side-effects of pimozide are concentration-dependent [301], the FDA recommends CYP2D6 genotyping [299]. In poor CYP2D6 metabolizers, the maximum dose should not exceed 0.05 mg/kg/day, and titration should be no faster than every 14 days [299].

Aripiprazole is an atypical neuroleptic that is approved by the FDA for the treatment of tics. It is a dopamine-serotonin partial agonist, acting as a partial D2, D3, and D4 receptor agonist, as well as a partial 5-HT1A, 5-HT2A, and 5-HT2C agonist [302]. A large multicenter, double-blind, randomized placebo-controlled study of children and adolescents with TS demonstrated that aripiprazole was superior to placebo in the reduction of tics with good tolerability [303]. Compared to pimozide, aripiprazole has a safer cardiovascular profile [297] and is generally quite well tolerated. The starting dose of aripiprazole is usually 1.25–2.5 mg/day, with gradual titration from 2.5 mg/day to 20 mg/day divided into two doses according to tic severity and tolerability [290,304].

Risperidone is frequently used for the treatment of tics, although it is not FDA-approved for this indication. It acts as a D2 and 5-HT2 receptor antagonist [305]. Risperidone may effectively reduce aggressive behavior as well as tic severity in patients with TS [306]. An open-label trial [305] and a randomized, double-blind, placebo-controlled trial [307] of patients with TS showed that risperidone reduces tic severity. A randomized, double-blind, parallel-group clinical trial of children with TS investigating the efficacy and tolerability of risperidone and clonidine demonstrated that risperidone was as effective as clonidine for the treatment of tic symptoms [308]. The adverse effects of risperidone include sedation, extrapyramidal symptoms (e.g., acute dystonic reactions, parkinsonism, and akathisia), orthostatic hypotension, hyperprolactinemia, gynecomastia, and weight gain. The starting dose is 0.25 mg daily, and it should be increased slowly every 5–7 days up to 0.25–4.0 mg daily [290].

Olanzapine is an atypical neuroleptic with a multiple-receptor-blocking profile including D1–D4, 5-HT, muscarinic, and histaminergic antagonists [309], and it has also been investigated for the treatment of tics. A clinical trial of children with TS showed that olanzapine was effective for tic suppression and aggression, but significant weight gain was noted [310]. Due to its important side-effects such as weight gain and sedation or drowsiness, olanzapine must be used cautiously in children [311]. Olanzapine is started at 2.5–5.0 mg/day and titrated every 5–7 days to a maximum of 30 mg/day [290].

Ziprasidone is an atypical neuroleptic and acts mainly as a 5-HT and D2 receptor antagonist and blocks other neurotransmitters [312]. A randomized, double-blind, placebo-controlled trial of children and adolescents with TS showed that ziprasidone reduced tics significantly and was tolerated well [313]. Unlike other atypical neuroleptics, ziprasidone does not appear to cause weight gain and extrapyramidal symptoms but is associated with a dose-dependent QTc interval prolongation [314]. Thus, patients with cardiac arrhythmia risk should avoid ziprasidone therapy. Ziprasidone is started at 5–10 mg/day and titrated slowly every week to a maximum of 40 mg/day [290].

Ecopipam is a selective D1 receptor agonist and has drawn attention as a treatment for tics. A randomized, placebo-controlled crossover study of children and adolescents with TS showed that ecopipam reduced tics significantly and was well tolerated [315]. Adverse events included gastrointestinal symptoms, decreased appetite, fatigue somnolence, headache, insomnia, rash, and nasopharyngitis, but no serious side effects were reported.

Compared to neuroleptics that block dopamine receptors, dopamine-depleting agents that deplete presynaptic dopamine by blocking the vesicular monoamine transporter type 2 (VMAT2) are safer with little or no risk of TD and have been used in the treatment of movement disorders such as chorea, TD, and tics [316]. Tetrabenazine is an FDA-approved medicine for the treatment of chorea associated with Huntington’s disease. Open-label clinical studies of patients with TS showed that tetrabenazine improved their tics and TS-related symptoms [317,318]. Side effects of tetrabenazine include drowsiness, sleepiness, akathisia, parkinsonism, and depression, which can be controlled by adjusting the dose [319]. Tetrabenazine is usually reserved for patients with severe tics who did not respond to or cannot tolerate other tic medicines. Importantly, tetrabenazine carries a black box warning regarding possible deterioration of an already present depression and should be used cautiously [319]. Although open-label studies suggested benefit of VMAT2 inhibitors for tics [320,321,322], controlled trials of two tetrabenazine derivatives in TS unfortunately failed to show efficacy. An open-label study of children and adolescents with TS failed to show benefit of valbenazine, which is a purified parent drug of the (+)-α-isomer of tetrabenazine [323]. The other trial failed to show significant benefit of deutetrabenazine, a deuterated from of tetrabenazine (ARTISTS1; NCT03452943) (https://clinicaltrials.gov/ct2/show/results/NCT03452943 (accessed on 01 June 2021)).

#### 7.3.3. Other Treatment

Aside from oral pharmacological treatment, botulinum toxin injections have been used in the treatment of spasticity and movement disorders [324]. Botulinum toxin’s mechanism of action is to inhibit the release of neurotransmitters (e.g., acetylcholine) from the presynaptic nerve terminal by cleaving soluble N-ethylmaleimide-sensitive factor attachment protein receptor proteins, which are essential for presynaptic vesicle fusion [325]. An injection of botulinum toxin into muscles that cause tic movements improves tic symptoms but also reduces the premonitory urge [326,327]. Similarly, injection into the vocal cords also improves vocal tics and the premonitory urge [328]. Treatment with botulinum toxin should be considered in patients with severe self-injurious motor tics (e.g., repetitive cervical extension) to prevent the progression of disabling myelopathy [329].

Deep brain stimulation (DBS) is a neurosurgical procedure to implant a device called a neurostimulator to deliver electrical stimulation to a targeted brain region and may be a promising treatment for patients with tics [330]. Reported targets for DBS include the thalamus, globus pallidus internus, globus pallidus externus, anterior limb of the internal capsule, and nucleus accumbens [331]. The International Deep Brain Stimulation Database and Registry demonstrated that the overall adverse event rate was 35.4% including intracranial hemorrhage (1.3%), infection (3.2%), and lead explantation (0.6%), and stimulation-induced side effects such as dysarthria (6.3%) and paresthesia (8.2%) [330]. Some work has found improvement in over 50% of patients [332,333,334], whereas other reports suggest that the likelihood of benefit may vary as much as 9‒82% [335]. Further studies are needed to identify the optimum target, benefits and risks, indications, timing for the procedure, and stimulation parameters [336].

Although hospital admission due to tics is very rare, about 5% of patients with TS who were referred to a movement disorder clinic exhibited life-threatening tics, known as “malignant TS” [332]. Life-threatening and dangerous self-injurious tic behaviors such as hyperextension of the neck are indications for hospital admission [332]. If patients have suicidal ideation, psychiatric hospital admission will often be warranted. Treatment for malignant TS is challenging. The first-tier and second-tier tic medicines are used, but may not be effective. Behavioral therapy or CBIT may be used with some benefit. For cases refractory to behavior therapy and medication, DBS of the globus pallidus interna or centromedial thalamus may be an option. Abrupt withdrawal of neuroleptics or clonazepam may lead to severe, disabling, and continuous tics, referred to as “tic status.” Tic status may interfere with activities and sleep, can be refractory to tic-suppressing medicines, and may require sedation with propofol and midazolam [337].

## 8. Differential Diagnosis

Various movements may resemble tics, from the common (e.g., habits, stereotypies, and mannerisms) to the abnormal (e.g., compulsion, chorea, dystonia, and myoclonus). To distinguish such movements from tics, one can confirm the presence of the premonitory urge, suppressibility, and suggestibility, as well as the phenomenology and timing of the movements.

Habits (and body-focused repetitive behaviors) are movements elicited by environmental stimuli or contexts and not performed to obtain future outcomes (e.g., nail biting, picking, thumb sucking, and hair twirling) [338]. Stereotypies are described as repetitive, patterned, non-purposeful movements (e.g., body rocking, tapping, hand flapping, arm waving) that stop with distraction or calling the child’s name, are frequently seen in early childhood, and may persist until adolescence with gradual reduction in frequency and duration [339]. They can occur in otherwise healthy children and in children with ADHD, OCD, anxiety, and autistic spectrum disorder [340]. Tics and stereotypies may coexist but may be differentiated by certain features. Stereotypies usually begin before the age of 3 years, which is earlier than the onset of tics, and they tend to improve during childhood [341]. Compared to tics, stereotypies are longer in duration, rhythmic, and less variable in type, location, and severity over time [341]. The premonitory urge is not seen in patients with stereotypies. Stereotypies often provide a comforting, enjoyable, and pleasing experience for children, in contrast to the discomfort and distress of tics [340]. Reassurance and psychoeducation are usually appropriate for stereotypies, but in some patients they can reduce quality of life, and in these patients, behavioral therapies such as habit-reversal training or response interruption and redirection can reduce the severity and frequency of stereotypies, while pharmacotherapy is usually not effective [342]. The use of an instructional DVD as a home-based, parent-administered behavioral therapy has been shown to reduce stereotypies by 15–24% [343].

Mannerisms are repetitive and unusual habits or gestures unique to the individual [344,345]. Mannerisms can be seen not only in healthy individuals but also in patients with schizophrenia with delusions [346]. Unlike tics, mannerisms may be goal-directed (e.g., performing a ritualistic action for luck) [345,346]. Mannerisms do not require treatment.

Compulsions are repetitive behaviors (e.g., hand washing) or mental acts (e.g., praying, counting) that are performed in response to an obsession or to prevent distress [5]. Some complex motor tics such as touching, tapping, and knocking resemble compulsions. Clinically, compulsions are associated with specific, sometimes ritualistic rules, or are performed in response to an obsession to reduce anxiety, distress, or discomfort [347]. Unlike patients with tics, those with compulsions do not experience premonitory sensations. Compulsions and complex tics may both occur in patients but distinguishing between them can be important to guiding treatment of the most bothersome symptoms. Compulsions improve with treatments for OCD, such as behavior therapy or serotonin reuptake inhibitors.

Chorea is characterized by brief, abrupt, irregular, unpredictable, purposeless, non-stereotyped movements flowing randomly from one part of the body to another [348]. Damage or dysfunction of the interconnection between the motor cortical areas and the basal ganglia, including the caudate nucleus, putamen, globus pallidus interna and externa, and associated structures such as the subthalamic nucleus and substantia nigra, leads to deficient inhibition to the thalamus and excessive thalamocortical motor facilitation, resulting in chorea movements [349]. The etiologies of acute chorea include autoimmune disease, infection, vascular disease, mitochondrial disease, toxins, and functional disease and those of chronic chorea include genetic, metabolic, and vascular diseases [350]. Differentiating between chorea and tics is aided by a detailed history of the disease onset, progression, and associated symptoms. Chorea may be voluntarily suppressible [37], but unlike tics, patients with chorea do not have premonitory urges. Compared to chorea, tic movements are usually more patterned and are repeated in a predictable and stereotypical manner.

Dystonia is a movement disorder characterized by involuntary sustained muscle contractions that produce abnormal postures or repetitive movements [351]. Electromyography (EMG) has shown that tonic agonist and antagonist muscle contractions occur during dystonia [352]. The etiology of dystonia is not fully understood, but structural abnormalities in the basal ganglia, cerebellum, cortex, brainstem, and thalamus, as well as neurotransmitter diseases that can affect dopaminergic dysfunction, have been suggested [353]. The most distinctive features of dystonia are sustained twisting movements, a sensory phenomenon called sensory trick (i.e., an internally generated, specific voluntary movement aimed at ameliorating the dystonia), task specificity, and directionality (e.g., alternates between quick jerking movements in one direction and slower movements in the opposite direction) [351,354]. Unlike tics, dystonia is not preceded by premonitory urges.

Myoclonus is a movement disorder characterized by brief, sudden, involuntary muscle jerks. It arises from all levels of the nervous system, including the cortical and subcortical areas, brainstem, spinal cord, and peripheral areas, and causes muscle contractions (positive myoclonus) or brief inhibition during sustained posture (negative myoclonus) [355]. Tics and myoclonus may be visibly indistinguishable, but some clinical features are useful to distinguish them. Myoclonus is very brief (traditionally <200 ms on EMG), is non-suppressible, may be unpredictable in terms of location and timing, and is not associated with a premonitory urge [356].

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
