# Peer review of "A Comprehensive Review of Tic Disorders in Children"

_jcm, 2021, doi:10.3390/jcm10112479_

Round 1

Reviewer 1 Report

Thank you for this very interesting and important article which gives a thorough review of the existing literature.

I have a few comments:

Concerning the effect of treatment (especially therapeutic treatment), a description of the literature analysing predictors and moderators of treatment outcome could have been included.

Longterm outcome of treatment procedures are not discussed.

Page 8, line 381: PANS: definition is "Pediatric Acute-onset Neuropsychiatric Syndrome"

Otherwise, only few minor spelling mistakes.

Reviewer 2 Report

General remark.

Except providing current, extensive knowledge, the review article should formulate unresolved items, address controversial topics on the subject (TS, Tourette Syndrome), include authors’ opinion on these problems supported by current and past literature.  However, I did not find these statements in the article.

Detailed remarks.

I have found some controversial opinion expressed by the authors in the manuscript which should be explained.

Line 461 – this is very controversial opinion because there is statement in product characteristics that methylphenidate is contraindicated in patients with tics. Despite literature data, the clinicians must follow this recommendation and enjoining. Additionally we have learned from the clinics, there are some patients who did have exacerbation of tics following administration of psychostimulants.   The authors cannot it leave without any comment not to confuse the readers.

Pharmacological treatment. You defined first- and second-line medicines. If I understood correctly, you prefer e.g. levetiracetam or baclofen (included in the first-line drug paragraph) over aripiprazole (included in second-line medicines paragraph). Despite that both, levetiracetam and baclofen, failed to become effective in placebo-controlled trials. Please address European and American recommendations to explain such a selection, and provide other references to suport your thesis.

DBS. It is very important treatment but only one sentence regarding the efficacy was given that we can expect ca. 50% reduction in tic intensity.  The readers can conclude that DBS is so effective as in Parkinson’s Disease but there are some research data that DBS is ineffective in treatment of refractory tics in TS (please provide). I strongly insist to give more profound description of DBS results in TS and more balanced conclusions. Please provide for anatomical targets electrodes were implemented.

Clinical course of co-morbid disorders. Line 177 – ADHD persists into adulthood, line 429 – age-related decline in ADHD (plus OCD) symptoms.  These is not in line. The literature data say that e.g ADHD symptoms decline in up to 65% of pts in contrast to OCD and depression symptoms that increase over time. Please provide clinical course of most important comorbid psychiatric disorders (ADHD, OCD, anxiety, depresion, ODD/CD), the rate in TS, if this rate is higher than in general, age- and sex- population,  if these disorders increase or decline in symptoms over time.

Comorbidity. Please summarize in short paragagraph, which of them share genetic background with TS, which are genetically linked but not having shared genetics with TS, which are not genetically linked to TS. Divide psychopathology into externalizing and internalizing  disorders. 

Developmental disorder. Many pts with TS have the delay in reaching the motor/language milestones during the development. If this is more frequently seen compared to general population? Please proviode a short paragraph on it.

Genetics. You have provided complete set of genes involved in TS pathogenesis but it is only partial true. You should give some clear massage for general clinicians/practisioners, e.g. TS is polygenic, rare variants, especially in non-coding regions,  play an important role with great impact of enviromental factors. Additionally, please try to explain why we still are not able to learn the genetic cause of TS despite extensive research over the world.

PANDAS. I cannot understand why you can diagnose PANDAS if the patient had tics before (line 350). This points to  primary tic disorder exacerbated by infection that is not rarely seen in clinical practice. Further, temporal relations must be confirmed with clinical GAS symptoms  - how you define temporal relations, one-two-more weeks? Based on literature or your own clinical experience. And finally, do you believe PANDAS exists? PANS does, for sure.

You wrote that comorbidities pay more attention to clinicians than tics themselves (lines 425-8). I cannot agree. In psychiatric clinics – yes, but not in neurological setting. We pay attention to tics  first of all. This depends on where the patients have been reffered, those with more severe psychopathology to psychiatrist, those with more severe tics to neurologist.

 Paragraph 7.2. Other treatment is  efficient in open studies due to placebo effect.  Comment it.

Differential diagnosis. Myoclonus vs tics: visibly indistinguishable, I agree. Do you think that EMG can differentiate myoclonus from tics? Please take into consideration motor artifacts during EMG resulted from violent movements.

Please determine a symptom hierarchy to differentiate habits vs stereotypies vs tics (important for general clinicians). Consider negative influence on daily living from stereotypies (see DSM 5 criteria), not habits, and consistent movement pattern for stereotypies vs waxing and waning for tics.

The challenge is to differantiate  stereotypies from tics in ASD. Any suggestions?

I really cannot understand the title of the article. Why other movement disorders in children? Manuscript concerns only tics.

Round 2

Reviewer 2 Report

The revised version of manuscript addresses all my concerns and previous remarks